# A Novel Epilepsy Detection Method Based on Feature Extraction by Deep Autoencoder on EEG Signal

**DOI:** 10.3390/ijerph192215110

**Published:** 2022-11-16

**Authors:** Xiaojie Huang, Xiangtao Sun, Lijun Zhang, Tong Zhu, Hao Yang, Qingsong Xiong, Lijie Feng

**Affiliations:** 1School of Basic Medical Sciences, Anhui Medical University, Hefei 230032, China; 2Institute of Biopharmaceuticals, Anhui Medical University, Hefei 230032, China; 3Department of Disaster Mitigation for Structures, Tongji University, Shanghai 200092, China

**Keywords:** EEG recognition, epilepsy detection, autoencoder, machine learning, model interpretability

## Abstract

Electroencephalogram (EEG) signals are the gold standard tool for detecting epileptic seizures. Long-term EEG signal monitoring is a promising method to realize real-time and automatic epilepsy detection with the assistance of computer-aided techniques and the Internet of Medical Things (IoMT) devices. Machine learning (ML) algorithms combined with advanced feature extraction methods have been widely explored to precisely recognize EEG signals, while among which, little attention has been paid to high computing costs and severe information losses. The lack of model interpretability also impedes the wider application and deeper understanding of ML methods in epilepsy detection. In this research, a novel feature extraction method based on an autoencoder (AE) is proposed in the time domain. The architecture and mechanism are elaborated. In this method, specified features are defined and calculated on the basis of signal reconstruction quantification of the AE. The EEG recognition is performed to validate the effectiveness of the proposed detection method, and the prediction accuracy reached 97%. To further investigate the superiority of the proposed AE-based feature extraction method, a widely used feature extraction method, PCA, is allocated for comparison. In order to understand the underlying working mechanism, permutation importance and SHapley Additive exPlanations (SHAP) are conducted for model interpretability, and the results further confirm the reasonability and effectiveness of the extracted features by AE reconstruction. With high computing efficiency in the time domain and an extensively satisfactory accuracy, the proposed epilepsy detection method exhibits great superiority and potential in almost real-time and automatic epilepsy monitoring.

## 1. Introduction

Epilepsy seizure, characterized by abnormal neuronal activities in the human brain [1], is a common disease in the brain‘s nervous system and seriously threatens normal life and even the safety of millions of patients around the world [2]. Epileptic patients may suffer from uncontrolled jerking movements, loss of consciousness, and other painful symptoms, which may result in unexpected death if patients are not treated in time [3]. Electroencephalogram (EEG) signals have been proven to be the golden tool to detect and analyze epilepsy [4]. With the development of the wearable EEG signal recording system [5,6], computer-aided techniques, and the Internet of Medical Things (IoMT) devices, continuously monitoring the EEG signal of epileptic patients is a promising method to provide real-time epilepsy monitoring to avoid catastrophic accidents. The common method for extracting physiological and disease information from the EEG waveforms is manually checking the long-term recordings and analyzing the wave morphology by experienced neurologists or doctors, which is time-consuming, labor-intensive, experience-dependent and inefficient for real-time and long-term epilepsy detection. Automatically recognizing EEG signals with high efficiency is the bottleneck for real-time automatic epilepsy detection.

Recently, many machine learning (ML) methods, including support vector machine (SVM) [7], decision tree [8], naïve Bayes, and KNN [9], have been explored for automatically detecting epilepsy with the feature extracted from EEG signals in the time domain [10], frequency domain [11] and time–frequency domain [12]. He et al. [7] compared the performances of support vector machine (SVM) and gradient boosting decision tree (GBDT) on automatic EEG signal recognition using empirical mode decomposition (EMD)-based time domain features, a nonlinear power spectrum density (PSD)-based feature. The results demonstrated that the GBDT achieved better performance. Albaqami et al. [8] utilized wavelet packet decomposition (WPD) to decompose the EEG signals and extracted the statistical features from the selected coefficients. In this research, the features aggregation algorithm was employed to reduce the dimension, and the extracted features were fed into a GDBT classifier. The results showed that the proposed method based on WPD and GBDT classifier exhibited higher accuracy and sensitivity than existing techniques. Wang et al. [9] combined the weighted KNN classifier and Bray Curtis distance to realize automatic epilepsy detection and the results suggested that the proposed method improved the prediction accuracy and reduced the false alarm rate. Xu et al. [13] investigated the GBDT classifier and nonlinear features including entropy, sample entropy, permutation entropy, spectral entropy, and wavelet entropy for epilepsy prediction, and the results showed that the proposed method could precisely recognize the two categories of EEG signal. 

It can be clearly seen that automatic epilepsy detection based on ML algorithms can be divided into two procedures, namely feature extraction, and classifier design. Many studies have been dedicated to exploring stylish classification algorithms and complicated feature extraction methods to improve prediction accuracy, while few of them took the computational cost into consideration, which impedes those methods for real-time epilepsy detection based on a wearable device with limited computing power [14]. 

The features of EEG data could be extracted in the time domain, frequency domain, and time–frequency domain. To be specific, the features extracted from raw data in the time domain (e.g., amplitude, entropy, etc.) possess the advantages of easy implementation and high computing efficiency, while the traditional time domain methods may lose significant epileptic details contained in the EEG signals. The characteristics of EEG signals can also be described in the frequency domain using signal transformation, and the most widely used frequency domain feature extraction approach is PSD. The PSDs of different kinds of EEG signals (health, interictal, and ictal signals) from an EEG dataset of the University of Bonn were demonstrated to be significantly different from each other [15]. Obtaining features individually from the time domain or frequency domain is inadequate in fully mining the information. Many studies transfer the one-dimensional raw EEG data into two-dimensional time–frequency images to extract the hidden features of the EEG signals for ML-based signal recognition. Ozdemir et al. [16] applied a Fourier-based synchrosqueezing transform (SST) with high resolution in time–frequency domain to convert the raw data into time–frequency images and a convolutional neural network (CNN) was utilized for signal classification. The proposed SST-based CNN method predicted epilepsy seizures with satisfactory accuracy. Although the features extracted from the frequency and time–frequency domain could provide abundant knowledge about epilepsy, those methods require a transformation process. Moreover, the convolution processing of time–frequency images for feature extraction and further image classification requires high computing power, especially GPU devices. Those factors make the feature extraction methods in frequency and time–frequency domains incongruous for real-time epilepsy detection. Therefore, the question of how to extract significant information from EEG signals in time domain with high efficiency is open-ended and yet to be answered. 

Many successful applications of ML methods based on EEG signal recognition have demonstrated their great potential and superiority in automatic epilepsy detection. However, model interpretability, which is critical for training convergence and generalization of the model [17], is habitually ignored by existing publications, which hinders wider applications of ML methods in epilepsy detection. Enhancing the model interpretability is significant for promoting the user’s understanding and predictive performance of ML-based epilepsy detection.

In this research, a novel epilepsy detection method based on feature extraction using a deep autoencoder (AE) without time–frequency transformation is proposed driven by EEG signals. AE-based features are employed as inputs to three typical ML classifiers to validate the effectiveness and superiority of the proposed AE-based time domain feature extraction method in epilepsy detection. Principal component analysis (PCA)-based features are fed into the same classifiers for comparison. Both feature distribution analysis and model interpretability (permutation importance analysis and SHapley Additive exPlanations (SHAP) method) were conducted to understand the underlying mechanism and to explain the superiority of the AE-based feature extraction method. This study aims to advance the ML-based method in real-time epilepsy detection using EEG signals. 

## 2. Methodology

In the proposed method, a certain type of EEG signal is utilized for decoding and encoding in the training of the AE model. Thereupon, when other types of signals are used as inputs as a test, the output reconstruction errors will exhibit a corresponding discrepancy since the trained AUE model is learned from a specified type of EEG characteristics. On this basis, the sensitive features can be extracted through the quantification of reconstruction error, which is further allocated for downstream classifier training.

The proposed AE is first utilized to reconstruct the EEG signals in the time domain. The indicators for the signal reconstruction quantification error are employed as the epilepsy-sensitive features, and the distribution patterns of the obtained features are presented to preliminarily investigate the effectiveness of AE-based extracted features. The obtained features are fed into three typical classifiers as predictor training for epilepsy detection. Finally, model interpretability analysis is conducted to figure out the effects of the different parameters on the model and to explain the superiority of the proposed method. The overall flowchart of this research is shown in Figure 1.

### 2.1. Dataset

The EEG dataset from the University of Bonn [15], a publicly available dataset, was utilized to verify the feasibility and effectiveness of the AE-based feature extraction method for automatic epilepsy detection. The dataset consists of five subsets, and each subset contains 100 single-channel EEG signals. The duration of each signal is 23.6 s and the sampling frequency is 173.6 Hz. Signals in sets A and B were recorded from five healthy volunteers with eyes open and eyes closed, respectively. In subsets C and D, the EEG data were measured from five epileptic patients during interictal intervals. The data in subset (E) were obtained from the five patients when active seizures occurred. In this research, the signals from set A, set D, and set E in the dataset were selected and utilized to represent the health, interictal and ictal EEG signals. To improve the generality of downstream models, the raw EEG signals were first normalized into the range 0 to1. The typical normalized signal samples of the three subsets are shown in Figure 2.

### 2.2. AE

AE, constituting decoder and encoder, is a kind of unsupervised neural network for distilling the immanent characteristics obscured in the raw data. Specifically, the encoder would encode time series signal **x** as a latent space vector **h**, while the decoder reconstructs the date from the latent space vector. The process could be expressed as:**h** = *φ*_1_ (*W*_1_**x** + *b*_1_)(1)
(2)$$\hat{x}=\varphi_{2}(\hat{W}h+b_{2})$$
where *φ*_1_ and *φ*_2_ are the activation functions of the encoder and decoder, *W* and $\hat{W}$ are the weight matrix, $\hat{x}$ is the reconstructed signal from AE, and *b*_1_ and *b*_2_ are the bias between the original input and the reconstructed ones obtained by the AE.

The encoder optimizes the weight matrix and bias matrix through backpropagation to minimize the reconstructed error. Mean squared error (MSE) is the most common loss function and could be expressed as:(3)$$L(\theta )=\frac{1}{N}\sum_{i=1}^{N}{\left|\right|x^{\left(i\right)}-\hat{x}||}^{2}$$
where *N* is the dimension of the input, and **x** and $\hat{x}$ are the input and output of the *i*th iteration, respectively.

As shown in Figure 3, AE in this study consisted of a series of stacked hidden layers for reconstructing the EEG signals. Specifically, layers E1 to E3 are the encoding module for compressing the input data into the latent space vector. Layer D1 to D3 represent the decoding module, which reconstructs the input data from the latent space vector. Each hidden layer is made up of a convolutional layer and an activation layer (ReLU). The loss function is defined as the MSE, and the learning rate of 0.001 is employed in the Adam optimizer to update the network weight matrix based on the training data.

In this study, the AE, trained using EEG signals from the interictal condition, is well-prepared for calculating the epilepsy-sensitive features in the time domain when the monitored EEG data were inputted into the primed AE, which could then enhance the efficiency of feature extraction. Moreover, the AE has demonstrated a remarkable ability to preserve the essence of the input EEG signals while eliminating noise, which may make it a superior tool to maintain more information than the typical dimension reduction method (e.g., PCA). From the perspective of computational efficiency and information preservation, the AE is therefore explored to extract features from EEG signals in the time domain for the automatic and real-time detection of epilepsy.

Three typical features including MSE, original-to-reconstructed signal ratio (ORSR), and cosine similarity (CS) are respectively defined to quantify the signal reconstruction and are further used as epilepsy-sensitive features. MSE is the regular indicator to quantify the difference between two signals, and could be expressed as:(4)$$MSE=\frac{1}{n}\sum_{i=1}^{n}{\left(x_{i}-{\hat{x}}_{i}\right)}^{2}$$
where *n* is the length of the original data, the higher MSE value indicates a larger reconstructed error while the lower MSE value reveals a smaller reconstructed bias. MSE has demonstrated its effectiveness in measuring the signal difference in many fields [18] and is employed in this study.

Inspired by the signal-to-noise ratio, ORSR is defined as the ratio between the amplitude of the original and reconstructed signals, and could be expressed as:(5)$$ORSR=10\log10\frac{\sum_{i-1}^{n}x_{i}^{2}}{\sum_{i-1}^{n}{\hat{x}}_{i}^{2}}$$

CS could quantify the signal similarity by mapping the data in the vector domain and calculating the cosine value between vectors. Higher cosine values suggest a lower similarity and vice versa. The cosine value of two vectors could be obtained as:(6)$$\cos\alpha =\frac{mn}{\left|m||n\right|}$$

The discrete expression form of the equation could be expressed as:(7)$$\cos(x,y)=\frac{\sum_{i=1}^{n}x_{i}{\hat{x}}_{i}}{\sqrt{\sum_{i=1}^{n}x_{i}^{2}}\sqrt{\sum_{i=1}^{n}{\hat{x}}_{i}^{2}}}$$

The mentioned indicators including MSE, ORSR, and CS are integrated as the input for representing the deep features of the EEG data and the automatic epilepsy detection in a data-driven manner.

### 2.3. Supervised Machine Learning Classifiers for Epilepsy Detection

With the rapid development of computational power, many studies have been conducted to achieve the goal of automatic epilepsy detection based on ML algorithms. To demonstrate the effectiveness and superiority of the AE-based features in EEG signal recognition, three widely used ML classifiers including the random forest (RF) classifier, the AdaBoost classifier, and the Gradient Boosting classifier are employed in this research. The working principles of the three classifiers are summarized as follows.

#### 2.3.1. RF Classifier

RF classifier [19] is based on an integrated algorithm of decision tree theory and consists of multiple decision tree classifiers. Each individual decision tree generates a prediction result, and the final prediction is performed by implementing the major voting method, as shown in Figure 4. The RF classifier has been widely utilized to classify EEG signals for evaluating the stages of sleep and diagnosing sleep problems [20] and identifying landscape perception [21] due to the good balance between execution time and reliability. Therefore, RF is employed to classify the EEG signals for the evaluation of the performance of the features extracted from the AE.

#### 2.3.2. AdaBoost Classifier

AdaBoost, namely adaptive boosting, possesses the adaptive characteristic of focusing on well-predicted samples. AdaBoost could be regarded as a training framework that could employ any classification algorithm and improve performance through a combinational approach. The basic principle and procedures are shown in Figure 5.

#### 2.3.3. Gradient Boosting Classifier

The Gradient Boosting classifier is a prevalent classification algorithm based on decision tree. This classifier has been widely utilized in EEG signal recognition [7,22] because of the advantages it has in reducing the over-fitting problem and has high prediction accuracy. The workflow of the Gradient Boosting classifier is shown in Figure 6.

The three typical ML classifiers were utilized to investigate the performance of the AE-based features in EEG signal recognition and automatic epilepsy detection.

## 3. Experimental Validation

The proposed AE-based feature extraction was verified on the EEG dataset of the University of Bonn, and the PCA-based analysis was also conducted as a comparison.

### 3.1. AE-Based Features Extraction

Considering the accessibility and representativeness of epileptic signals, subset D (EEG signals from the interictal condition) was selected to train the AE for feature extraction in this study. The time domain waveforms of the reconstructed EEG signals and the original ones from the subset A (healthy condition), subset D (interictal condition), and subset E (ictal condition) of the EEG dataset from the University of Bonn are shown in Figure 7. 

It can be seen that the reconstructed signals of the healthy and ictal conditions critically differ from the original ones, while the reconstructed interictal EEG signals highly coincide with the original waveform. This phenomenon is reasonable because the AE is trained by the interictal EEG signal, and the EEG signals obtained from the healthy condition and the epileptic attack period significantly differ from interictal EEG signals. The differences in signal reconstruction in the different kinds of EEG signals preliminarily demonstrate the high sensitivity of the proposed AE method.

The three indicators (CS, MSE, and ORSR) quantifying the signal reconstruction were utilized as the epilepsy-sensitive features. The characteristics of the three features were explored using the distribution pattern plots, as shown in Figure 8. It is obvious that the three indicators globally exhibit an obvious clustering pattern. The CS of the interictal state presented a higher value while the MSE and ORSR exhibited lower levels. The satisfactory distribution patterns of the features demonstrate the potential of utilizing the three indexes for EEG signal classification and further epilepsy detection. 

### 3.2. PCA-Based Features Extraction 

The essence of extracting features from raw signals is in reducing dimension while remaining the important information. Therefore, the widely used dimension reduction method, PCA, is utilized to extract the features and provide a comparison for the AE.

PCA utilizes statistical tools to represent the original data in a lower-dimensional space [23]. In this research, the obtained principal components (PCs) were presented for the preliminary exploration of the validity of PCA (with the number of components as 10) in feature extraction. Figure 9 depicts the first three PCs which were selected and normalized for the clear presentation of their distribution pattern. The selected PCs exhibited inconspicuous clustering patterns, which demonstrated that the PCs extracted from the PCA may be inadequate in extracting the features required for representing the EEG signals from different conditions.

## 4. Predication Accuracy and Model Interpretability

The obtained features from the AE were fed into the three ML classifiers (RF classifier, AdaBoost classifier, and the Gradient Boosting classifier) to investigate the performance of the AE-based features in automatic epilepsy detection. The same analysis was conducted on PCA to provide a comparison. Permutation importance analysis and the SHAP method were implemented to interpret the model and explain the superiority of the AE-based features in epilepsy detection. 

### 4.1. AE-Based Features for Epilepsy Detection

To confirm the effectiveness of AE in feature extraction, the parameters of MSE, ORSR, and CS were combined as the input of the aforementioned ML classifiers to recognize epilepsy. In each classifier, 75% of the input features were randomly selected as the training set, while the rest of the input data were used as the testing set. 

A confusion matrix, a specific layout for visualizing the classification performance of supervised learning methods, was utilized to quantitively demonstrate the efficiency of the extracted features by the AE in each classifier, and the results are shown in Figure 10. 

The confusion matrices of training outcomes are presented in Figure 10a, and it can be seen that the three classifiers exhibit a training accuracy of almost 100%, which suggests a reasonable relationship between the AE-based features and epilepsy. The testing performance of the AE features in the three classifiers is shown in Figure 10b, and the prediction accuracy was 97%. Only two health EEG signals are wrongly categorized into an interictal and ictal state in all three classifiers, while other signals were successfully recognized. From the perspective of safety, a false alarm by taking a healthy EEG signal as an epileptic signal is more acceptable than missing the ictal symbol. The classification results of the three ML tools demonstrated the high accuracy and stable performances of the AE-based features in epilepsy detection. 

### 4.2. PCA-Based Features for Epilepsy Detection

The PCs obtained by PCA were combined and then fed into the same three classifiers to provide a comparison for the proposed AE-based feature extraction method in epilepsy detection. 

The confusion matrices of the training outcomes are presented in Figure 11a. The RF classifier and Gradient Boosting classifier reached an accuracy of 100%, while the AdaBoost classifier exhibited a poor accuracy of 73%, which shows that the performances of the PCA-based features are unstable for different classifiers. The testing accuracies of the PCs in the three classifiers are shown in Figure 11b. the Gradient Boosting classifier achieved the highest accuracy of 72%, while the accuracies of the RF classifier and Ada Boosting classifier were 69% and 52%, respectively. 

The accuracies of the AE-based features and PCA-based features are presented in Table 1. With the PCs as the input, the predicting accuracies of the three classifiers were critically lower than the features extracted by the proposed AE. Moreover, not only were healthy EEG signals categorized as epileptic symbols, but the interictal and ictal signals were also wrongly classified as healthy signals when the PCs were utilized as the input of the ML methods. The results revealed the poor accuracy and stability of the PCA-based features for EEG signal recognition and further epilepsy detection. 

### 4.3. Model Interpretability

Most studies utilizing the ML methods in EEG signal classification and epilepsy detection end up with the presentation of classification accuracy, while a few of them conduct a model interpretability analysis, which may lead to an unsatisfactory disease treatment protocol and even threaten the patients’ safety. Model interpretability is the connection between human beings and the model, which not only describes the trained model accurately but can also be comprehensible to the users [24]. SHapley Additive exPlanations (SHAP) [25], a framework to explain the black-box model, is consistent with human intuition and is becoming increasingly popular in the interpretation of models in medical and social fields [26]. Permutation importance is another powerful tool to understand the effects of each feature on the trained model [27].

To fully understand the underlying mechanism of classification models using the selected features, the two abovementioned model interpretability methods were conducted to analyze the influences of the AE-based features on the fitted model. The relative importance of each AE-based feature in the RF classifier was obtained, as shown in Figure 12. It can be seen that CS and MSE play relatively important roles, while the ORSR exhibited less of a contribution in the classification model. SHAP feature importance was obtained by averaging the absolute SHAP values per AE-based feature, and sorting them by decreasing importance, as shown in Figure 13. The presenting results were analogous to the permutation importance, and the CS contributed the most of all the three classifiers, while the ORSR had the lowest effect on the models. Both the relative importance and the SHAP value demonstrated that the effect of the three indicators on the trained model was comparative in magnitude extent, which further verified the efficacy of using the extracted features for training the downstream classifiers.

## 5. Discussions

The feature extraction of EEG signals with high efficiency is the critical step in automatic epilepsy detection using ML-based methods. Most studies have paid much attention to the improvement of prediction accuracy using sophisticated signal processing methods, and few of them focused on obtaining the features with a high computation efficiency and a high degree of information preservation. In this research, the deep AE was proposed to extract the features in the time domain to avoid the transformation process and to ensure computing efficiency. Meanwhile, AE-based features preserved more significant EEG signal information than the PCs obtained from PCA. The obtained parameters were then utilized to train the classification models and predict epilepsy. The feature distribution and interpretability model were then performed to advance the understanding of the users in the trained classification model. 

A limitation of this study might have been that the analysis was mainly conducted on the EEG dataset from the University of Bonn, and the generalization of the AE-based feature extraction method for automatic epilepsy detection needs to be further studied. Three features including CS, ORSR, and MSE were extracted in the time domain based on the proposed AE, and more prevalent time domain features of the EEG signals (e.g., amplitude, zero-crossing rate, etc.) need to be investigated in epilepsy detection. 

## 6. Conclusions

In this research, a deep AE was proposed to extract the features of EEG signals in the time domain for providing an efficient feature extraction algorithm for real-time epilepsy detection. The obtained features were fed into three classical classifiers (namely the RF classifier, AdaBoost classifier, and Gradient Boosting classifier) for automatic epilepsy detection. Meanwhile, the performance of the proposed method was compared with PCA, and the results demonstrated that the AE-based features achieved a prediction accuracy of 97%, which was much higher than that of the features extracted from PCA. Moreover, the AE-based features exhibited higher robustness and stability than the PCA-based features when utilizing the different classifiers. The distribution pattern of the features and model interpretability analysis further verified the reasonability of the proposed AE-based features for epilepsy. This novel feature extraction method of EEG signals based on AE has great potential in improving computing efficiency and promoting the applications of ML-based automatic epilepsy detection.

## Figures and Tables

**Figure 1 ijerph-19-15110-f001:**
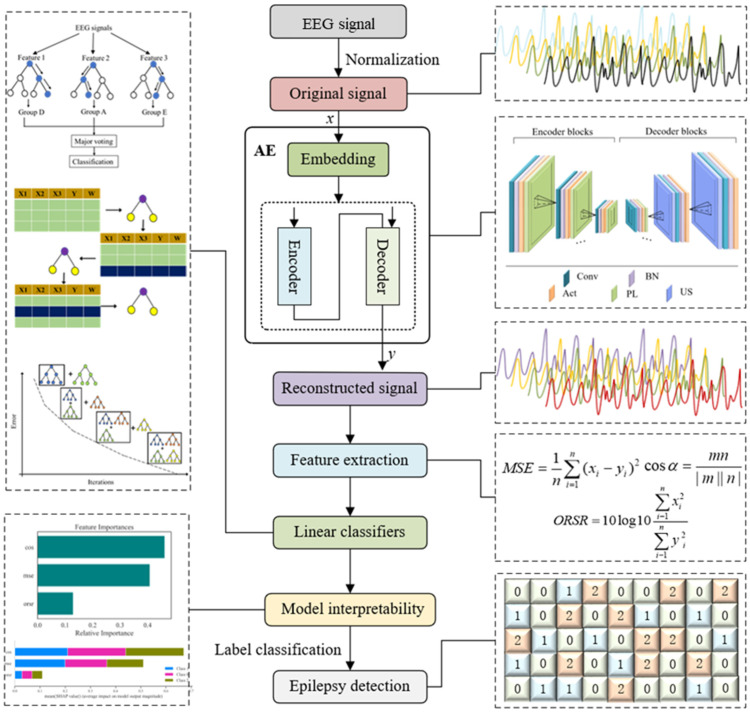
Flowchart of this research.

**Figure 2 ijerph-19-15110-f002:**
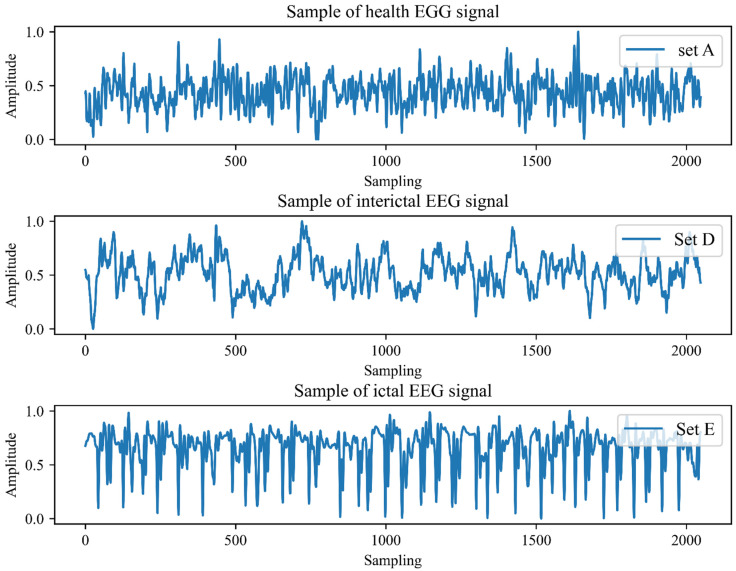
The normalized waveforms in the time domain for set A, set D, and set E.

**Figure 3 ijerph-19-15110-f003:**
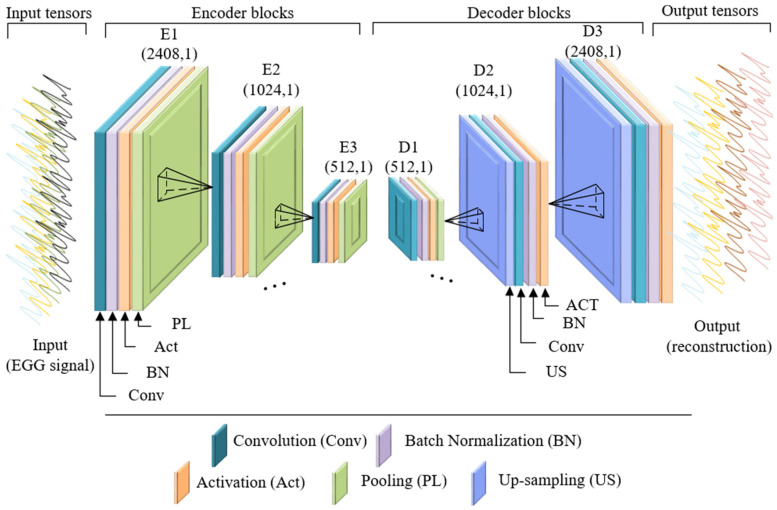
Architecture of the proposed deep AE.

**Figure 4 ijerph-19-15110-f004:**
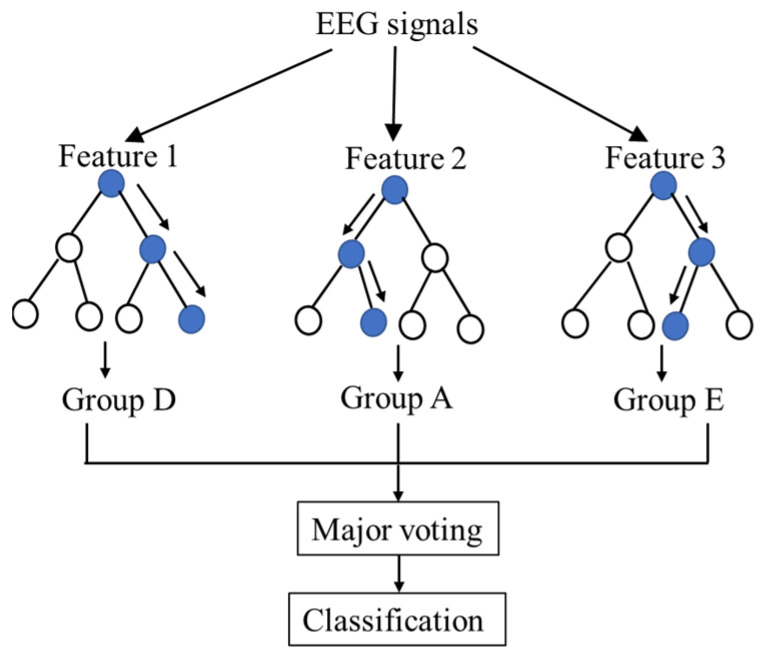
Schematic procedures of the RF classifier.

**Figure 5 ijerph-19-15110-f005:**
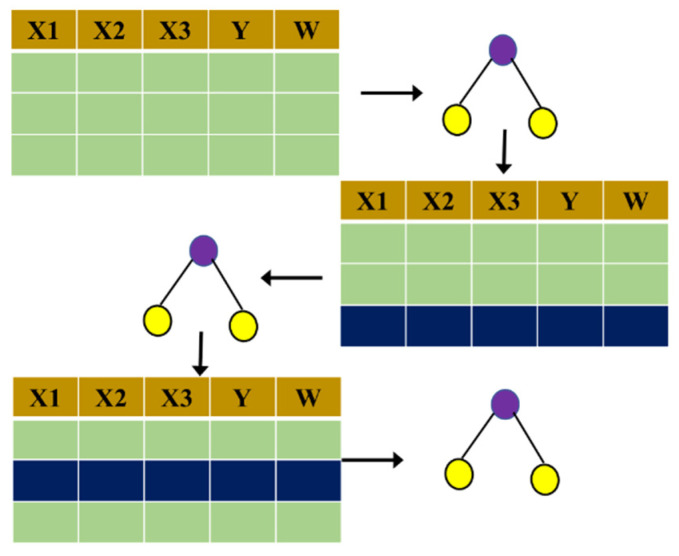
Schematic procedures of the AdaBoost classifier.

**Figure 6 ijerph-19-15110-f006:**
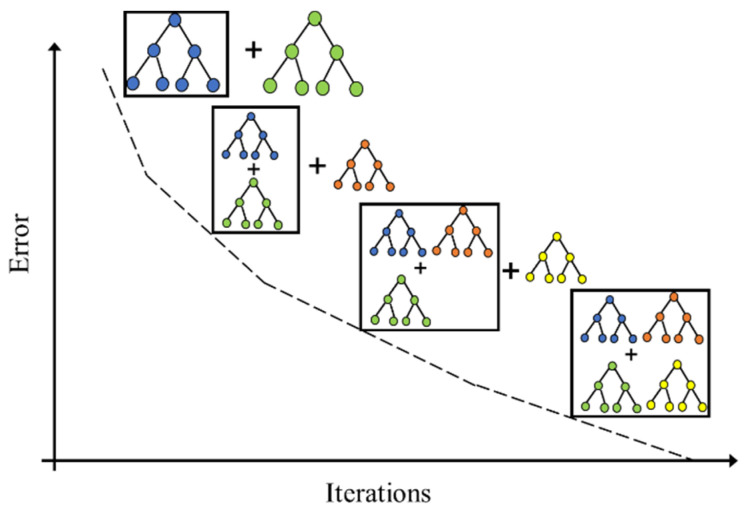
Schematic procedures of the Gradient Boosting classifier.

**Figure 7 ijerph-19-15110-f007:**
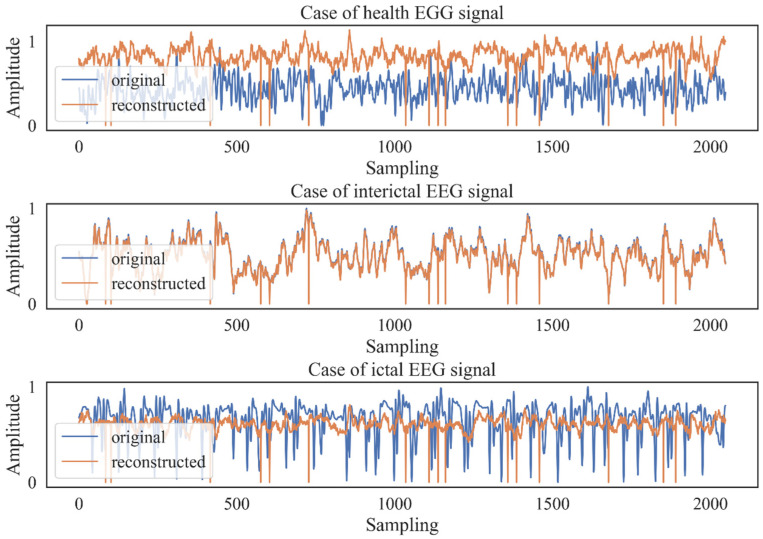
Reconstruction results for the healthy, interictal, and ictal EEG signals.

**Figure 8 ijerph-19-15110-f008:**
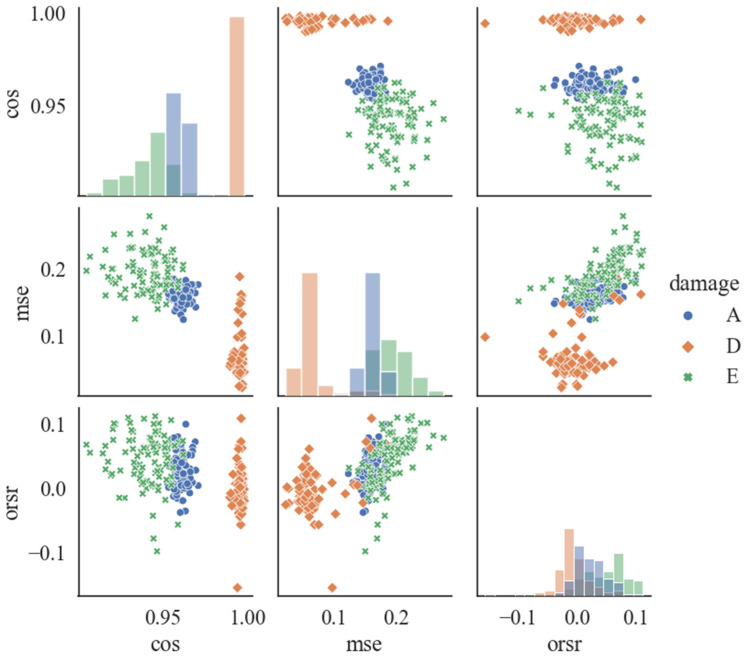
Distribution patterns for the AE-based features.

**Figure 9 ijerph-19-15110-f009:**
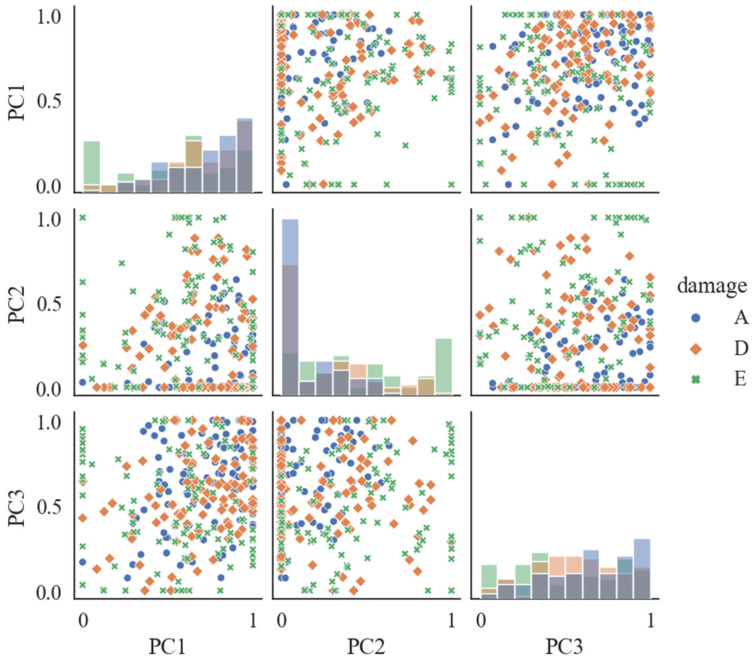
Distribution patterns of the PCA-based features.

**Figure 10 ijerph-19-15110-f010:**
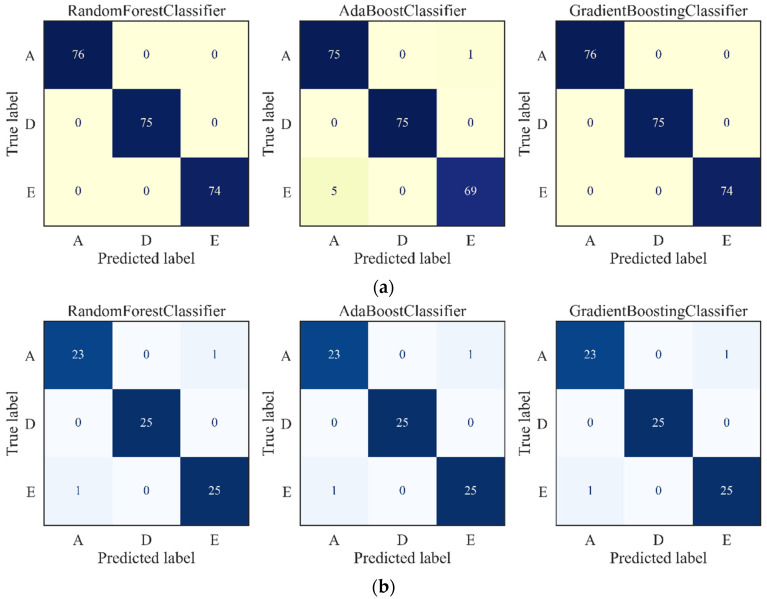
Prediction results using the AE-based features as inputs: (**a**) Confusion matrix of training, (**b**) confusion matrix of the test.

**Figure 11 ijerph-19-15110-f011:**
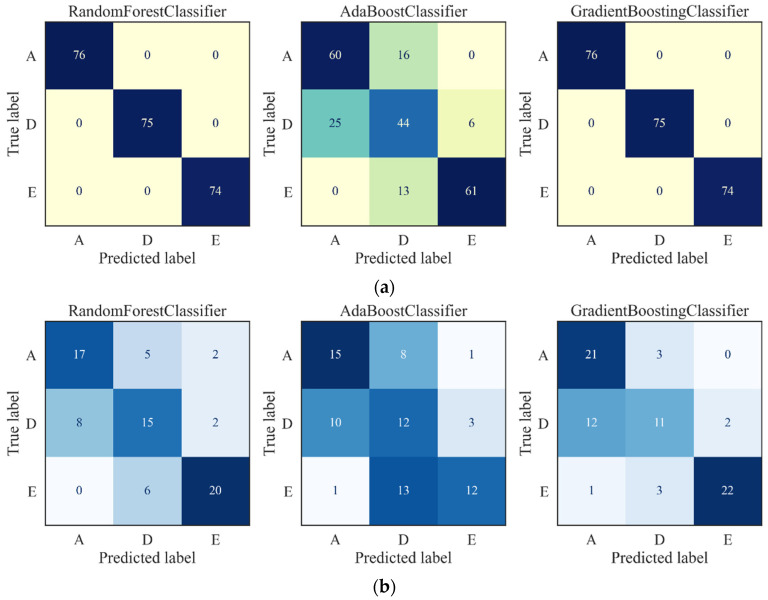
Prediction results using PCA-based features as inputs: (**a**) Confusion matrix of training, (**b**) confusion matrix of the test.

**Figure 12 ijerph-19-15110-f012:**
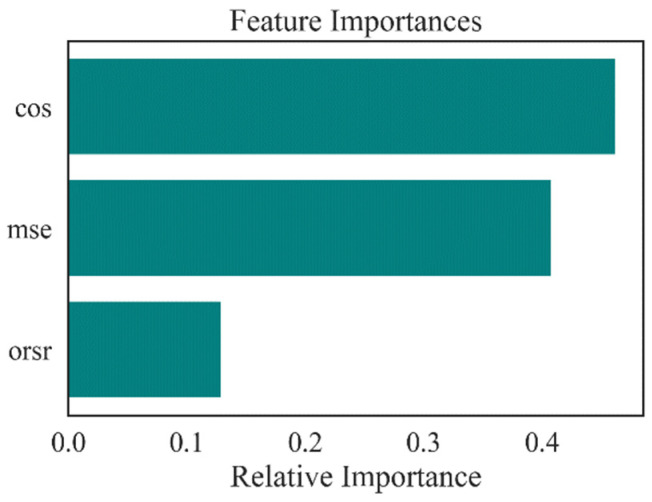
Permutation importance of the AE-based features.

**Figure 13 ijerph-19-15110-f013:**
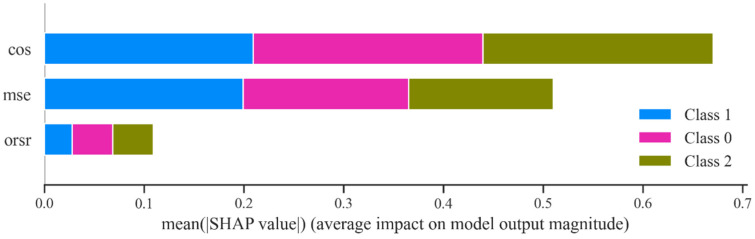
SHAP values of the AE-based features.

**Table 1 ijerph-19-15110-t001:** Accuracies of the AE-based and PCA-based features in the three classifiers.

	RF	AdaBoost	Gradient Boosting
Training accuracy of AE features	100%	97.3%	100%
Testing accuracy of AE features	97.3%	97.3%	97.3%
Training accuracy of PCA features	100%	73.3%	100%
Testing accuracy of PCA features	69.3%	52%	72%

## Data Availability

The university of Bonn EEG dataset can be found at the following website: http://epileptologie-bonn.de/cms/upload/workgroup/lehnertz/ (accessed on 15 November 2022).

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
