# Peer review of "A Novel Epilepsy Detection Method Based on Feature Extraction by Deep Autoencoder on EEG Signal"

_ijerph, 2022, doi:10.3390/ijerph192215110_

Round 1
Reviewer 1 Report
In this paper, the authors propose an epileptic classification using an autoencoder, what are the advantages of an autoencoder and why an autoencoder is appropriate for this case.
typo in Abstract: EEG --> EGG
Many of the illustrations in Figure 1 are illegible because the text/pictures are too small. It is recommended that illustrations be enlarged or omitted and presented when discussing related processes.
Typo in Figure 2, EEG--> EGG
The font size on lines 148-150, 154 and 155 should be the same as the body text
It is very difficult to compare the accuracy between methods when using a confusion matrix, you should use a table to make it easier for readers and provide measurements of sensitivity and specificity.
Reviewer 2 Report
In this paper the Authors present a new approach to epilepsy detection using AE.
Although the Authors indicate novelty of their approach in the paper title, I would rather consider this approach to be to a level similar to other approaches published in numerous papers where Authors use ANN of some kind to perform classification / clustering task. The main difference is the structure of the ANN which Authors of this paper decided to use. And here is where I have a bit of a problem with this paper. Because, as long as I appreciate a very thorough methodology description, experiment design, comparison of the obtained results with PCA method, then I can't find any justification of WHY the Authors chose the Autoencoder type of ANN to solve this specific problem. I have seen several publications which use Autoencoder ANN for features extraction, including other than medicine domains, but usually there is also a clear explanation where the gains making this approach better than other come from. In this paper the Authors do not explain neither, WHAT made them use the AE ANN, nor where the improvement comes from. And this is my main criticism over this paper. If such things are not thoroughly explained, then one can make an assumption that the success of the Authors is achieved by an accident.
Other than that, everything is fine with this paper. This is a well-structured and well-designed study having (almost) all the elements which constitute a solid research publication. Furthermore, it is all written in good and comprehensible English.
I have not found any significant flaws in the paper, the only one which likely is a mistake is eq.4, where the MSE is not squared. So, once it appears the equation is indeed wrong, I would request the Authors to make a correction and the paper is ready for publication.
